# Selective Sweeps in the Austrian Turopolje and Other Commercial Pig Populations

**DOI:** 10.3390/ani13243749

**Published:** 2023-12-05

**Authors:** Farzad Atrian-Afiani, Beate Berger, Christian Draxl, Johann Sölkner, Gábor Mészáros

**Affiliations:** 1Institute of Livestock Sciences (NUWI), University of Natural Resources and Life Sciences, Vienna, 1180 Vienna, Austria; farzad.atrian-afiani@boku.ac.at (F.A.-A.); johann.soelkner@boku.ac.at (J.S.); 2Institut für Biologische Landwirtschaft und Biodiversität der Nutztiere, HBLFA Raumberg-Gumpenstein 2, 4600 Thalheim bei Wels, Austria; beate.berger@raumberg-gumpenstein.at; 3Österreichische Schweineprüfanstalt GmbH, 2004 Streitdorf, Austria; christian.draxl@smpa.at

**Keywords:** selection signatures, genomic analysis, muscular development genes

## Abstract

**Simple Summary:**

In our study, our primary aim was to identify signatures of selection in Turopolje pigs, a local breed, and four commercial pig breeds (Large White, Landrace, Pietrain, and Duroc). We conducted an extensive analysis of genetic data from 485 pigs, examining 54,075 specific genetic markers. Our focus was on genes that have undergone positive selection pressures. Consistent with previous research, we identified two key genes, *PTBP2* and *DPYD*, as consistently influenced by selection in all the pig breeds we studied. These genes play essential roles in muscular development, a critical factor for pigs and other species. Additionally, in the Large White breed, we discovered several genes associated with traits such as body weight, production efficiency, meat quality, and more. In the Duroc breed, we detected different genes linked to similar traits. Our findings provide valuable insights into the genetic changes driven by selective pressures in local and commercial pig breeds. This knowledge can enhance our understanding of pig breeding mechanisms and contribute to more efficient and sustainable pig farming practices. Our study underscores the significance of employing various genomic methodologies to identify genetic signatures of selection, fulfilling the primary aim of our research, and offers important insights into pig breed diversity.

**Abstract:**

The goal of our study was to identify signatures of selection in the Turopolje pigs and other commercial pig breeds. We conducted a comprehensive analysis of five datasets, including one local pig breed (Turopolje) and four commercial pig breeds (Large White, Landrace, Pietrain, and Duroc), using strict quality control measures. Our final dataset consisted of 485 individuals and 54,075 single nucleotide polymorphisms (SNPs). To detect selection signatures within these pig breeds, we utilized the XP-EHH and XP-nSL methodologies, which allowed us to identify candidate genes that have been subject to positive selection. Our analysis consistently highlighted the *PTBP2* and *DPYD* genes as commonly targeted by selection in the Turopolje breed. *DPYD* is associated with muscular development in pigs and other species and *PTBP2* emerges as one of the potential genes linked to seminal characteristics. Furthermore, in the Large White breed, a number of genes were detected with the two methods, such as *ATP1A1*, *CASQ2*, *CD2*, *IGSF3*, *MAB21L3*, *NHLH2*, *SLC22A15*, *VANGL1*. In the Duroc breed, a different set of genes was detected, such as *ARSB*, *BHMT*, *BHMT2*, *DMGDH*, *JMY*. The function of these genes was related to body weight, production efficiency and meat quality, average daily gain, and other similar traits. Overall, our results have identified a number of genomic regions that are under selective pressure between local and commercial pig breeds. This information can help to improve our understanding of the mechanisms underlying pig breeding, and ultimately contribute to the development of more efficient and sustainable pig production practices. Our study highlights the power of using multiple genomic methodologies to detect genetic signatures of selection, and provides important insights into the genetic diversity of pig breeds.

## 1. Introduction

Studying selective sweeps in domesticated animals is a valuable tool for identifying genomic regions that have been influenced by natural and artificial selection. Such studies can uncover important mutations, genes, and pathways associated with economically significant traits [1]. Over the years, commercial pig breeding programs have been developed to improve desirable traits, such as growth rate, meat quality, and disease resistance, in order to meet the increasing demand for pig products. Selection strategies within breeds and also crossbreeding are commonly used to achieve these goals [2]. These adaptations can result from both hard and soft selective pressures. Hard selective pressures involve the complete removal of deleterious alleles or the fixation of advantageous alleles, leading to a rapid and strong selective sweep. On the other hand, soft selective pressures involve the selection of multiple alleles with smaller effects, resulting in a more gradual and less pronounced selective sweep [3]. However, there are concerns that the intense selection pressure in commercial pig breeding programs may also lead to unintended consequences, such as a decrease in genetic diversity and increased susceptibility to disease [4]. Positive selection tends to favor a small number of highly productive animals, which can result in the loss of rare genetic variants that may be important for adaptation to changing environments or resistance to diseases.

Understanding the genetic mechanisms underlying these selective pressures can provide valuable insights into the evolutionary history of pigs and inform breeding strategies for improving pig health and productivity. Selective sweeps are important evolutionary events that contribute to the adaptation of populations to changing environments, and many methods are available to study the genetic mechanisms underlying these events. One such method is cross-population extended haplotype homozygosity (XP-EHH), an extension of the EHH method. This is a haplotype-based approach that measures the difference in haplotype length between two populations and identifies genomic regions that have undergone recent positive selection [5]. XP-EHH has been successfully used to detect selective sweeps in a variety of species, including humans [6], sheep [7], and pigs [8]. Another method that can be used to capture genomic patterns resulting from local selective sweeps is cross-population number of segregating sites by length (XP-NSL), which is an extension of the haplotype-based statistic (nSL). This method is useful for performing a genomic scan to identify regions of local adaptation by comparing haplotype patterns between two populations. XP-nSL has been shown to have good power in detecting both ongoing and recently completed hard and soft sweeps, making it a valuable tool for studying the genetic mechanisms underlying adaptation to different environments [9].

The XP-nSL method is designed to identify a selection pressure that has recently become prevalent in a population. When positive selection is in progress, or a beneficial mutation has recently become fixed in a population, there is a reduction in genetic diversity near the adaptive allele. Consequently, the observation of extended homozygous haplotypes at a high frequency is anticipated [9]. The capability to specifically target recent selection pressures proves particularly valuable in analyzing the Turopolje breed, given that the Austrian population of this breed was established only in the 1990s.

Integrating these two methods can provide us with a greater confidence in identifying genomic regions that have undergone positive selection. Overall, understanding the genetic basis of desirable traits in domesticated animals is essential for improving breeding programs, and the detection of selection signatures can play an important role in achieving this goal. Therefore, our study aimed to identify genomic regions showing evidence of recent positive selection, comparing commercial breeds with the Austrian Turopolje breed. The aim of the study was to gain insights into the evolutionary and biological mechanisms shaping the genetic characteristics of the Turopolje, and the other breeds.

## 2. Materials and Methods

### 2.1. Data

The present study leveraged genomic data from Austrian Turopolje and commercial pig populations. The commercial dataset comprised Large White (82 individuals), Landrace (76 individuals), Pietrain (69 individuals), and Duroc (74 individuals), while the local population (Turopolje) consisted of 184 individuals.

Turopolje is a local lard-type pig, developed as a typical pasture pig in swampy surroundings (Sava valley in Croatia). The breed history in Austria is connected to the rescue of four animals at the time of the war in the former Yugoslavia in the 1990s. Just four of these animals, with a later import of two additional boars, were the basis of the Austrian local population [10]. The pigs are smaller than modern high-yielding breeds, with a much lower percentage of lean meat (about 40% vs. 60% and more in modern breeds). The lard is of very high quality and used for traditional air-cured ham and other special products. The Turopolje samples were taken between 2018 and 2022, at the time the animals were registered in the herd book as breeding animals. Similarly, the genotypes of the commercial breeds originate from the respective active populations, with samples taken in 2022.

We conducted quality control using PLINK with the following parameters; only autosomal variants were retained, variants with less than 10% missing data (--mind 0.1) and individuals with less than 10% missing genotypic data were retained (--geno 0.1). After conducting quality control measures, a total of 54,075 single nucleotide polymorphisms (SNPs) were retained for analysis, with 485 individuals included in the final dataset. The genotyping rate for the dataset was 0.996.

### 2.2. Methods

In this study, two methods were used to identify regions under selection in the genomes of commercial and local pigs: cross-population extended haplotype homozygosity (XP-EHH) and cross-population number of segregating sites by length (XP-nSL). These methods were implemented in the software Selscan v.2.080 [11].

To perform haplotype phasing, Beagle v.5.4 [12] was used. Two approaches were used to determine the selection signatures with both methods: (1) each commercial population was compared with the Turopolje breed separately; (2) then all populations were combined into one group, and jointly compared to Turopolje.

Normalization of both methods scores across all chromosomes was performed using the “norm” program, distributed along with Selscan. Normalized XP-EHH scores were considered as the signature of selection.

XP-nSL statistics were used to detect local adaptations by comparing haplotype patterns between two populations around the same allele of interest. XP-nSL was designed to identify genomic regions involved in local adaptation between two populations. Building upon the nSL method used for within-population analysis, XP-nSL compares haplotype sets from different populations, calculating statistics based on ancestral and derived alleles. Computation involves assessing the similarity of consecutive sites within haplotypes, followed by normalization across the genome. This method has the power to detect both ongoing and recently completed hard and soft sweeps. In the XP-nSL method, when two populations are compared, a positive score suggests evidence of a hard or soft sweep in population 1, while negative scores indicate similar evidence in population 2 [9]. For this study, commercial breeds were identified as population one and independently compared with the Turopolje breed, designated as population two.

In total, 68 windows were analyzed using these methods to identify regions under selection in the genomes of commercial and local pigs.

## 3. Results

Our research revealed that commercial breeds had multiple regions under selection, as shown in Figure 1, Figure 2, Figure 3, Figure 4 and Figure 5 for Large White, Duroc, Pietrain, Landrace, and the entire population, respectively. Each figure illustrates the distribution of normalized XP-EHH and XP-nSL scores along the genomic position for pairwise comparisons of pig breeds. Positive scores indicate selection pressure on commercial breeds, while negative scores signify selection pressure on local breeds. Notably, in all pairwise comparisons, *PTBP2* and *DPYD* were consistently identified by both methods as candidate genes exhibiting selection signatures in Turopolje (Figure 1, Figure 2, Figure 3, Figure 4 and Figure 5). In our study, xp-EHH and xp-nSL detected 133 and 70 genes, respectively, as candidates for signatures of selection. Candidate genes were detected in the top 1% of XP-EHH and XP-nSL values (the top 1% XP-EHH and XP-nSL values are shown above and below the red and blue lines in Figure 1, Figure 2, Figure 3, Figure 4 and Figure 5). These selected regions are dispersed across different chromosomes, as detailed in Table 1. Notably, a significant portion of the positive selection signals is concentrated on chromosomes 1 and 4, highlighting their particular importance in our observations.

The fifteen common genomic regions between the two methods suggest the presence of positive selection in those areas. Within the Turopolje breed, XP-EHH and XP-nSL analyses revealed the presence of *POU4F1*, *OBI1*, *KLF4*, *DPYD*, *PTBP2*, *DOCK5*, *GNRH1*, *KCTD9*, *CDCA2*, and *EBF2* for XP-EHH, and *PTBP2*, *DPYD*, *MACROD2*, *KIF16B* for XP-nSL. Notably, both methods identified *PTBP2* and *DPYD*, which are situated on chromosome 4 (Table 1).

In the Large White breed a number of genes were detected by both selection signature methods used in this study, such as: *ATP1A1*, *CASQ2*, *CD2*, *IGSF3*, *MAB21L3*, *NHLH2*, *SLC22A15*, *VANGL1* (Table 2). In the Duroc breed, a distinct set of genes showed similarity between the two methods in our study, including *ARSB*, *BHMT*, *BHMT2*, *DMGDH*, and *JMY* (Table 2). For a comprehensive overview of additional genes detected by both methods across all commercial and local breeds, please refer to (Table 1).

## 4. Discussion

The candidate regions we have identified are spread out across various chromosomes. This distribution suggests that both natural selection and random mating may have played a partial role in shaping the genomes of these breeds [13]. In our study, we identified 15 common genomic regions that exhibited evidence of positive selection when analyzed with two distinct methods. These regions were specifically localized to chromosomes 1 and 4, as detailed in (Table 1 and Table 2). Notably, our analysis utilizing the XP-EHH and XP-nSL approaches consistently pinpointed the *PTBP2* and *DPYD* genes as recurrently favored targets of selection in the Turopolje breed, thereby emphasizing the robustness and reliability of these findings. The obtained results aligned with our expectations, as the focus of most breeding programs has been on muscular development in Turopolje breed. This alignment is attributed to the fact that the *DPYD* gene is associated with muscular traits in pigs. Furthermore, a multitude of genes exhibited positive selection in commercial breeds when compared to the Turopolje breed, encompassing *CD2*, *IGSF3*, *ATP1A1*, *MAB21L3*, *SLC22A15*, *NHLH2*, *CASQ2*, and *VANGL1* in Large Whites, as well as *ARSB*, *BHMT*, *BHMT2*, *DMGDH*, and *JMY* in the Duroc breed. While certain genes were detected independently by each method (see Table 1), our focus in this study centers on the common genes identified by both approaches. Below, we explored a detailed discussion of each individual gene.

A study by Santana et al. [14] suggested that genetic variation in the *DPYD* gene could influence muscle growth and development in Nellore cattle, and might be associated with the rib eye area as a measure of muscle quantity. The authors found significant associations between a single nucleotide polymorphism (SNP) in the *DPYD* gene and the rib eye area in Nellore cattle, providing further support for the potential relevance of this gene in pig breeding. Studies showed that the *DPYD* is one of the candidate genes in the Duroc pig population that were detected within or nearby significant SNPs related to intramuscular fat (IMF) metabolism. In addition, the *DPYD* is related to muscle contraction, muscle system process, developmental process, and sphingolipid signaling pathway [15]. Furthermore, the studies showed that *PTBP2* emerges as one of the potential genes linked to seminal characteristics in Duroc pigs. Specifically, *PTBP2*, along with *STRA8*, stands out as a noteworthy candidate gene concerning sperm progressive motility. The regulation of *PTBP2* splicing, a key process involving the polypyrimidine tract binding protein 2, plays a crucial role in facilitating communication between germ cells and Sertoli cells. [16].

In the Large White breed, several genes were identified using both selection signature methods, as previously mentioned in this study (Table 2). Further details are elaborated below. A study by Shanshan et al. [17] suggested that the *ATP1A1* gene is a protein-coding gene that encodes the alpha-1 subunit of the Na+/K+-ATPase enzyme, which is involved in the active transport of sodium and potassium ions across cell membranes. In addition, there is disrupted potassium ion homeostasis in the ciliary muscle in negative lens-induced myopia, and that this may be related to changes in the expression and activity of Na+/K+-ATPase, which is encoded by the *ATP1A1* gene. Specifically, they noted that there is decreased expression of *ATP1A1* at both the mRNA and protein levels in the ciliary muscles with negative lens induced myopia, as well as reduced Na+/K+-ATPase activity. These findings suggest that the disruption of potassium ion homeostasis may contribute to the development of myopia in these animals, and that changes in the expression and activity of *ATP1A1* may be involved in this process [17]. The *CASQ2* gene, as with the *DPYD*, revealed significant enrichment in KEGG pathways in relation to muscle contraction, muscle system process, developmental process, and the sphingolipid signaling pathway [15]. In investigating intramuscular fat (IMF) and pork quality, researchers identified sex-specific biomarkers related to IMF in pigs. Among these, the *CD2* molecule (*CD2*) emerged as a notable biomarker for high-IMF pigs, confirmed via qRT-PCR and shedding light on IMF deposition mechanisms for enhanced meat quality [18]. A genomic study by Fontanesi et al. [19] delved into the average daily gain of Italian Large White pigs, suggesting a connection between the *IGSF3* gene and variations in average daily gain.

The results underscored that the identified candidate genes primarily revolve around pig production and meat quality, indicating targeted selective pressures in commercial pig breeds. These genes appear to have undergone significant selection to enhance production efficiency and meat quality.

Our result also showed selection pressure on *NHLH2*, *SLC22A15*, and *VANGL1* in Large White pigs, which are located on chromosome 4. The *NHLH2* gene encodes a basic helix–loop–helix transcription factor implicated in regulating adult body weight and fertility, based on mouse knockout studies. Notably, two SNPs in the *NHLH2* gene have functional implications. One SNP within the 3′ untranslated region (3′UTR) results in increased mRNA instability and reduced protein levels when linked to luciferase mRNA. The other SNP in the coding region leads to an amino acid change affecting protein migration and secondary structure. One of these SNPs is associated with obesity in humans, while the other is prevalent among all individuals. These findings suggest a potential role for *NHLH2* in body weight regulation which warrants a further investigation into their influence on human weight and fertility phenotypes [20]. In the pig population studied, the gene *SLC22A15* was examined as part of the validation process using qRT-PCR to confirm the RNA-Seq results. *SLC22A15* was identified as one of the down-regulated genes. The qRT-PCR analysis of *SLC22A15*’s expression in pigs showed consistent results with the transcriptomic data, indicating that changes in *SLC22A15* expression due to different feeding patterns were reliably captured using both RNA-Seq and qRT-PCR methods. This highlights the role of *SLC22A15* in responding to varying feeding regimens within the pig population [21].

In another study by Bergamaschi et al. [22], results showed that the composition of the gut microbiome was influenced by the host’s genetics, and several SNPs were found to be associated with specific microbial taxa at different time points during the pigs’ growth. These SNPs were located within genomic regions that contained a total of 68 genes. One of these genes, *VANGL1*, located in a genomic region on SSC4, was identified as part of the Wnt signaling pathway. *VANGL1* was found to be highly expressed in gut tissues and has been linked to processes such as cell proliferation and turnover. Interestingly, the study highlighted that *VANGL1* was associated with specific microbial Operational Taxonomic Units (OTUs), particularly those classified as Anaerostipes. This finding suggested a connection between *VANGL1* and the abundance of certain gut microbes.

In the Duroc breed, a distinct set of genes showed similarity in our study between the two methods, including *ARSB*, *BHMT*, *BHMT2*, *DMGDH*, and *JMY*. Similar to our results, the study by Wang et al. [23] explored genetic diversity and selection signals in pig populations, particularly highlighting the genes *ARSB*, *DMGDH*, and *BHMT* within the supergene region. Using ROH analysis on data from American Duroc (AD) and Canadian Duroc (CD) pig populations were studied. AD pigs exhibited higher genetic diversity and lower inbreeding compared to CD pigs, likely due to distinct selective pressures. In the overlapping ROH hotspots in both populations, a significant missense mutation (rs81216249) was identified in the supergene region containing *ARSB*, *DMGDH*, and *BHMT* genes. This variant allele, originating from European pigs, was nearly fixed in Duroc pigs. Strong selection on this supergene was indicated, possibly contributing to improvements in body weight and length. These findings suggest the potential for the genetic enhancement of pigs using these genes as trait-related markers. They identified a significant genetic variant situated within the overlapping section of three genes: *ARSB*, *DMGDH*, and *BHMT*. Specifically, the variant manifested as a missense mutation within the *DMGDH* gene. Analysis of various genetic indicators pointed towards the positive selection acting upon this genetic region encompassing the three genes (*ARSB*-*DMGDH*-*BHMT*). This observation suggested that this region had undergone a process of favorable selection.

This observation strongly implied a history of favorable selection that shaped this genomic region. Notably, these findings mirror our own outcomes, where both XP-EHH and XP-nSL methodologies demonstrated a similar pattern of positive selection on these genes, as showcased in Figure 3. The results showed a significant genetic variant within the *ARSB*-*DMGDH*-*BHMT* gene cluster that underwent positive selection, as evidenced by various genetic indicators and the uniform genetic haplotype among Duroc pigs. This gene cluster’s influence on growth and fat deposition traits adds to its biological significance [23].

The gene *JMY* also has potential involvement in muscle development, along with other genes like *HOMER1*, as well as *ITGA1* and *RAB32*. These genes are associated with functions such as glycogen metabolism and mitochondrial dynamics, which are relevant to muscle health and development [24].

*BHMT2*, functioning as a methyl transferase, facilitates the transfer of a methyl group from betaine to homocysteine, resulting in the production of methionine. This biochemical process intricately interweaves with essential metabolic pathways involving dimethylglycine, betaine, and methionine. *BHMT2*’s influence extends to growth and metabolism, as its capability to catalyze methyl transfers contributes depth and intricacy to these metabolic interactions. Notably, the genetic correlation between betaine, dimethylglycine, and methionine levels and growth rates suggests that *BHMT2* plays a role in influencing growth potential [25].

The results of this study indicate that the candidate genes identified were primarily associated with pig production and meat quality, suggesting that selection pressure in commercial pig breeds has been largely focused on these traits. Specifically, these genes appear to have been under strong selective pressure in order to improve the production efficiency and meat quality of commercial pig breeds.

## 5. Conclusions

Our study identified several noteworthy genes targeted by the selection process, such as *DPYD* linked to muscle growth, and *PTBP2* emerges as one of the potential genes linked to seminal characteristics, significant in the Turopolje breed. In commercial breeds such as Large White and Duroc pigs, candidate genes, including *ATP1A1*, *CASQ2*, *CD2*, *IGSF3*, *NHLH2*, *SLC22A15*, and *VANGL1*, were demonstrated to have relevance to muscle functions, development, and fat metabolism. Furthermore, genes like *ARSB*, *BHMT*, *BHMT2*, *DMGDH*, and *JMY* indicated positive selection for growth traits. The consistent identification of genetic variants and positive selection signals across methodologies underscores the importance of these genes in shaping pig production traits. This information is crucial in understanding the genetic processes that contribute to differentiation in pig breeds thanks to a better understanding of natural and artificial selection processes.

## Figures and Tables

**Figure 1 animals-13-03749-f001:**
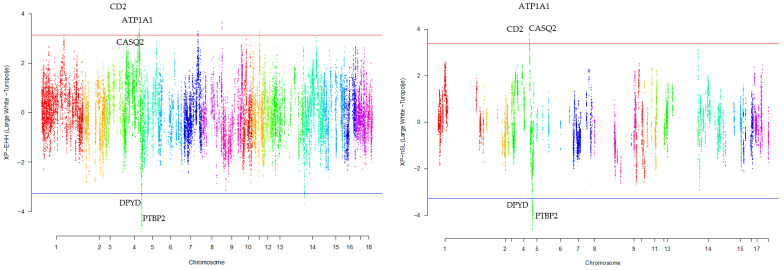
Genomic distribution of normalized cross-population extended haplotype homozygosity (XP-EHH) scores and XP-nSL in pairwise Large White and Turopolje (Top 1% values above and below red and blue lines).

**Figure 2 animals-13-03749-f002:**
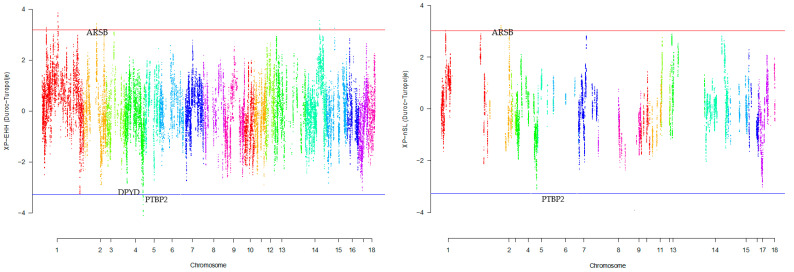
Genomic distribution of normalized cross-population extended haplotype homozygosity (XP-EHH) scores and XP-nSL in pairwise Duroc and Turopolje detected on 18 chromosomes (Top 1% values above and below red and blue lines).

**Figure 3 animals-13-03749-f003:**
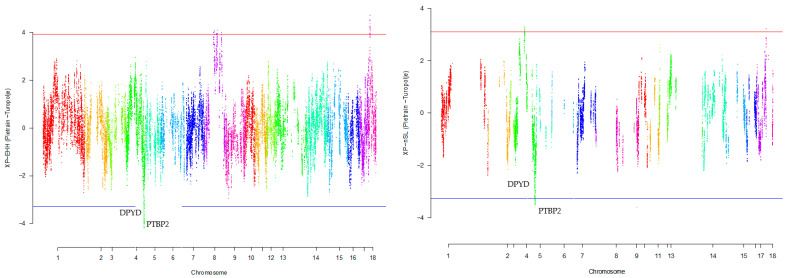
Genomic distribution of normalized cross-population extended haplotype homozygosity (XP-EHH) scores and XP-nSL in pairwise Pietrain and Turopolje detected on 18 chromosomes (Top 1% values above and below red and blue lines).

**Figure 4 animals-13-03749-f004:**
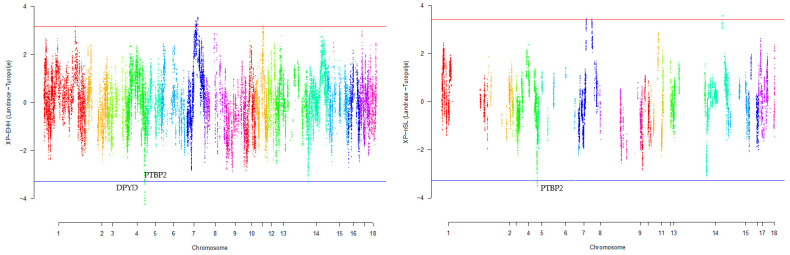
Genomic distribution of normalized cross-population extended haplotype homozygosity (XP-EHH) scores and XP-nSL in pairwise Landrace and Turopolje detected on 18 chromosomes (Top 1% values above and below red and blue lines).

**Figure 5 animals-13-03749-f005:**
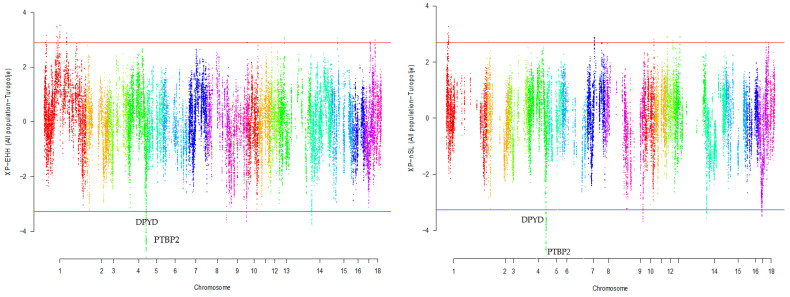
Genomic distribution of normalized cross-population extended haplotype homozygosity (XP-EHH) scores and XP-nSL in pairwise All population and Turopolje detected on 18 chromosomes (Top 1% values above and below red and blue lines).

**Table 1 animals-13-03749-t001:** Top one percent of genes detected by XP-EHH and XP-nSL methods.

Breed Names	Chr	Genomic Region	XP-EHH	Chr	Genomic Region	XP-nSL
Large white	1	147690556–148087719	*ZNF236*, *ZNF516*	4	103963345– 105048775	*CD2*, *IGSF3*, *ATP1A1*, *MAB21L3*, *SLC22A15*, *NHLH2*, *CASQ2*, *VANGL1*
	4	102827190–105048775	*WDR3*, *GDAP2*, *TENT5C*, *MAN1A2*, *CD2*, *IGSF3*, *ATP1A1*, *MAB21L3*, *SLC22A15*, *NHLH2*, *CASQ2*, *VANGL1*			
	7	95365939– 95930034	*RGS6*, *DPF3*			
	8	136963965– 137878333	*CFAP299*, *FGF5*, *PRDM8*, *ANTXR2*			
	11	51339629– 51560305	*RBM26*, *NDFIP2*			
Duroc	1	28566266 –107135517	*AHI1*, *MYB*, *HBS1L*, *ALDH8A1*, *MBD2*, *POLI*, *STARD6*, *C18orf54*, *DYNAP*, *ST8SIA3*, *ONECUT2*, *FECH*, *NARS1*, *MTFMT*, *SPG21*, *ANKDD1A*	1	27958270– 28302666	*PDE7B*
	2	87322934 –88089403	*LHFPL2*, *ARSB*, *DMGDH*, *BHMT2*, *BHMT*, *JMY*	2	87614963– 137175305	*ARSB*, *DMGDH*, *BHMT2*, *BHMT*, *JMY*, *HOMER1*, *TENT2*, *CMYA5*, *MTX3* *THBS4*, *SERINC5*, *C5orf15*, *VDAC1*, *TCF7*, *PPP2CA*, *CDKL3*, *UBE2B*, *CDKN2AIPNL*, *JADE2*, *SAR1B*, *SEC24A*, *CAMLG*, *DDX46*, *C5orf24*, *TXNDC15*, *CATSPER3*
	14	93978255– 94102060	*ZWINT*			
	15	55491087– 55681505	*DUSP4*, *TNKS*			
Pietrain	8	69771934– 116334013	*RASS*, *6CXCL8*, *AMCF-II*, *PPBP*, *CXCL2*, *MTHFD2L*, *EPGN*, *INPP4B*, *IL15*, *RNF150*, *TBC1D9*, *ELMOD2*, *MGAT4D*, *CLGN*, *SCOC* *PPA2*	18	59387007 –62455499	*ZFHX4*, *HNF4G*, *CRISPLD1*, *PI15*, *JPH1*, *LY96*, *TMEM70*, *UBE2W*, *STAU2*
	18	10268960– 12177267	*LUC7L2*, *FMC1*, *UBN2*, *TTC26*, *ZC3HAV1*, *ZC3HAV1L*, *KIAA1549*, *TMEM213*, *ATP6V0A4*, *SVOPL*, *TRIM24*, *AKR1D1*, *CREB3L2*, *DGKI*	4	9717294 –10234121	*TBXAS1*, *HIPK2*, *CLEC2L*, *KLRG2*
Landrace	1	206792212– 207495965	*CCDC171*, *PSIP1*, *SNAPC3*, *TTC39B*	14	49973938– 50184089	*IL16*, *STARD5* *TMC3*
	7	57223851– 74975357	*PEAK1*, *HMG20A*, *LINGO1*, *ODF3L1*, *CSPG4*, *SNX33*, *IMP3*, *SNUPN*, *PTPN9*, *UBL7*, *SEMA7A*, *CYP11A1*, *CCDC33*, *STRA6*, *ISLR*, *ISLR2*, *PML*, *STOML1*, *LOXL1*, *TBC1D21*, *INSYN1*, *NPTN*, *REC114*, *HCN4*, *NEO1*, *ADPGK*, *BBS4*, *EGLN3*, *NPAS3*, *AKAP6*, *ARHGAP5*, *STXBP6*, *GZMB*, *GZMH*, *SDR39U1*, *KHNYN*, *CBLN3*, *NYNRIN*, *NFATC4*, *RIPK3*, *ADCY4*, *LTB4R*	7	100691955 –102597121	*LIPN*, *LIPM*, *ANKRD22*, *STAMBPL1*, *ACTA2*, *FAS*, *CH25H*, *IFIT3*, *IFIT1*, *IFIT5*, *SLC16A12*, *PANK1*, *KIF20B*, *HTR7*, *RPP30*,
	11	50730428– 50786774	*POU4F1* *OBI1*			
Turopolje	1	248604820– 248609035	*KLF4*	4	119931414 121041802	*DPYD*, *PTBP2*
	4	119931414– 121041802	*DPYD*, *PTBP2*	17	22635284– 25186703	*MACROD2*, *KIF16B*
	14	9197293– 10025164	*DOCK5*, *GNRH1*, *KCTD9*, *CDCA2*, *EBF2*			

**Table 2 animals-13-03749-t002:** Common genes across different methods in pairwise comparison between commercial breeds and Turopolje.

Comparing Breed and Methods	Common Genes		
	XP-EHH	XP-nSL	Between XP-EHH and XP-nSL
Large White (Between two methods)	-	-	*DPYD*, *PTBP2*, *ATP1A1*, *CASQ2*, *CD2*, *IGSF3*, *MAB21L3*, *NHLH2 SLC22A15*, *VANGL1*
Large White and Duroc	*DPYD*, *PTBP2*	*PTBP2*	*PTBP2*
Large White and Pietrain	*DPYD*, *PTBP2*	*DPYD*, *PTBP2*	*DPYD*, *PTBP2*
Large White and Landrace	*DPYD*, *PTBP2*	*PTBP2*	*PTBP2*
Duroc (Between two methods)	-	-	*ARSB*, *BHMT*, *BHMT2*, *DMGDH*, *JMY*
Duroc and Pietrain	*DPYD*, *PTBP2*	*PTBP2*	*PTBP2*
Duroc and Landrace	*DPYD*, *PTBP2*	*PTBP2*	*PTBP2*
Pietrain (Between two methods)	-	-	*DPYD*, *PTBP2*
Pietrain and Landrace	*DPYD*, *PTBP2*	*PTBP2*	*DPYD*, *PTBP2*
Landrace (Between two methods)	-	-	*PTBP2*

## Data Availability

The genomic data resulted from a collaboration involving Boku University, the Institut für Biologische Landwirtschaft und Biodiversität der Nutztiere at HBLFA Raumberg-Gumpenstein, and Österreichische Schweineprüfanstalt GmbH.

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
