# Peer review of "Selective Sweeps in the Austrian Turopolje and Other Commercial Pig Populations"

_animals, 2023, doi:10.3390/ani13243749_

Round 1
Reviewer 1 Report
Comments and Suggestions for Authors
The submitted paper investigates the ability of two statistical methods developed for software analysis of haplotypes to predict regions under selection in the genome of commercial and local pigs. Selective sweeps have been identified and discussed for several genes, and their potential association with muscle metabolism and growth in commercial pigs is highlighted. On the whole, the analyses performed are correct, but the results and text are less clear and difficult to follow in some parts.
Some of the additional information/amendments listed below could further improve the text.
Row:
76: …that is…
81-82: Explain this statement in more detail.
89-91: This sentence is not clear, please rephrase it in a more understandable way.
93-102: Please give more details about the analysed data sets (breeding animals or fatteners, the date of sampling, etc.)
128: Figure 1. is missing
135: The title of Table 1 is missing.
Also, the results/discussion for the local breed seem to be missing.
145: …pig breeds (instead of pig species)
232: Which study?
Conclusion – just a note, I would be more careful with the claims in the conclusion as the study did not include phenotypic measurements. Also, the possibility of a further increase in the meatiness of today's commercial pigs is open to discussion.
Reference: 10., 14., 15., 16., 19. and 24. should be corrected and aligned with the rest of the list.
Author Response
Dear reviewer,
Thank you for the review of our paper. Here we provide a point-by-point response to your questions.
The submitted paper investigates the ability of two statistical methods developed for software analysis of haplotypes to predict regions under selection in the genome of commercial and local pigs. Selective sweeps have been identified and discussed for several genes, and their potential association with muscle metabolism and growth in commercial pigs is highlighted. On the whole, the analyses performed are correct, but the results and text are less clear and difficult to follow in some parts.
Some of the additional information/amendments listed below could further improve the text.
Row:
76: …that is…
Answer: Done.
81-82: Explain this statement in more detail.
Answer: Done.
89-91: This sentence is not clear, please rephrase it in a more understandable way.
Answer: Done.
93-102: Please give more details about the analysed data sets (breeding animals or fatteners, the date of sampling, etc.)
Answer: Details on the populations were added. All animals were the representatives of the recent population for the respective breed, with genotyping from live animals in or around 2022. There were no fatteners genotyped.
128: Figure 1. is missing
Answer: Done.
135: The title of Table 1 is missing.
Answer: Done, added.
Also, the results/discussion for the local breed seem to be missing.
Answer: Done, added.
145: …pig breeds (instead of pig species)
Answer: Done.
232: Which study?
Answer: Done, added.
Conclusion – just a note, I would be more careful with the claims in the conclusion as the study did not include phenotypic measurements. Also, the possibility of a further increase in the meatiness of today's commercial pigs is open to discussion.
Answer: Done, the text of conclusions was adjusted.
Reference: 10., 14., 15., 16., 19. and 24. should be corrected and aligned with the rest of the list.
Answer: Done.
Reviewer 2 Report
Comments and Suggestions for Authors
The topics of the manuscript are of interest for breeding practice. The results presented by the authors are an important addition to the existing state of knowledge in the field under discussion.
I consider the applied research methods to be correct. The selection of literature items is also correct.
In my opinion, the scientific work is worth publishing, however, it requires some corrections as to the presentation of the results.
I do not understand what the title of Table 1 is about? This is probably some kind of draft version?
I leave it to the Editor's discretion whether to describe its heading under Table 2 (although everything is given in the Material and methods chapter).
In addition, Fig. 2 and Fig. 3 are included in the manuscript and Fig. 1 is missing. Authors do not refer to the Figures presented.
Comments on the Quality of English Language
I rate the manuscript as linguistically correct.
Author Response
Dear reviewer,
Thank you for the review of our paper. Here we provide a point-by-point response to your questions.
The topics of the manuscript are of interest for breeding practice. The results presented by the authors are an important addition to the existing state of knowledge in the field under discussion.
I consider the applied research methods to be correct. The selection of literature items is also correct.
In my opinion, the scientific work is worth publishing, however, it requires some corrections as to the presentation of the results.
I do not understand what the title of Table 1 is about? This is probably some kind of draft version?
Answer: Done, corrected.
I leave it to the Editor's discretion whether to describe its heading under Table 2 (although everything is given in the Material and methods chapter).
Answer: Done.
In addition, Fig. 2 and Fig. 3 are included in the manuscript and Fig. 1 is missing. Authors do not refer to the Figures presented.
Answer: Done, corrected.
Reviewer 3 Report
Comments and Suggestions for Authors
The authors conducted a comprehensive analysis of selection signatures in Turopolje pigs and other commercial pig breeds using rigorous quality control measures and multiple datasets. They employed the XP-EHH and XP-nSL methodologies to detect selection signatures and identified candidate genes, such as PTBP2 and DPYD, which were consistently targeted by selection across all studied breeds and associated with muscular development in pigs and other species. Overall, this study reveals genomic regions under selective pressure between local and commercial pig breeds, providing valuable insights into pig breeding mechanisms and the development of more efficient and sustainable pig production practices.
However, I suggest the following improvements:
1.Gene names should be italicized to distinguish them from normal text.
2. What are the growth characteristics of Turopolje pigs? What are the main differences between Turopolje pigs and other commercial pig breeds? Please add this to the introduction section of the article.
3. What does the title of Table 1 mean? Please provide a concise title to replace it.
4. The title "chromosome number" in Table 1 should be changed to "chromosome."
5. Please add Figure 2 and Figure 3 to their respective positions in the Results section.
6. In the captions of Figure 2 and Figure 3, the meaning of the red lines should be explained. According to convention, the title of the figure should be placed below the figure.
7. Where are the 133 and 53 genes detected using the XP-EHH and XP-nSL methods, respectively?
8. If possible, please mark the positions of PTBP2 and DPYD on the Manhattan plot.
Author Response
Dear reviewer,
Thank you for the review of our paper. Here we provide a point-by-point response to your questions.
The authors conducted a comprehensive analysis of selection signatures in Turopolje pigs and other commercial pig breeds using rigorous quality control measures and multiple datasets. They employed the XP-EHH and XP-nSL methodologies to detect selection signatures and identified candidate genes, such as PTBP2 and DPYD, which were consistently targeted by selection across all studied breeds and associated with muscular development in pigs and other species. Overall, this study reveals genomic regions under selective pressure between local and commercial pig breeds, providing valuable insights into pig breeding mechanisms and the development of more efficient and sustainable pig production practices.
However, I suggest the following improvements:
1.Gene names should be italicized to distinguish them from normal text.
Answer: Done.
- What are the growth characteristics of Turopolje pigs? What are the main differences between Turopolje pigs and other commercial pig breeds? Please add this to the introduction section of the article.
Answer: Done.
- What does the title of Table 1 mean? Please provide a concise title to replace it.
Answer: Done.
- The title "chromosome number" in Table 1 should be changed to "chromosome."
Answer: Done.
- Please add Figure 2 and Figure 3 to their respective positions in the Results section.
Answer: Done.
- In the captions of Figure 2 and Figure 3, the meaning of the red lines should be explained. According to convention, the title of the figure should be placed below the figure.
Answer: Done.
- Where are the 133 and 53 genes detected using the XP-EHH and XP-nSL methods, respectively?
Answer: The edited number has been presented in a new table, displaying all detected genes.
- If possible, please mark the positions of PTBP2 and DPYD on the Manhattan plot.
Answer: Done.
Round 2
Reviewer 1 Report
Comments and Suggestions for Authors
I can confirm that the authors have addressed the issues raised in my comments and have improved the text, which now appears acceptable. The only detail that I think is still unclear is the statement in lines 106-107:
"The pigs are hardy, very good swimmers and are known even to dive for mussels and water plants. Characteristically they have webbed feet", which needs to be supported by evidence or references.
Author Response
Dear Reviewer,
Thank you for the review of our paper.
comment: The only detail that I think is still unclear is the statement in lines 106-107:
"The pigs are hardy, very good swimmers and are known even to dive for mussels and water plants. Characteristically they have webbed feet", which needs to be supported by evidence or references.
Answer: I have edited the paragraph per your feedback.
Reviewer 3 Report
Comments and Suggestions for Authors
The authors have made significant improvements to the manuscript. I have no more suggestions.
Author Response
Dear Reviewer,
Thank you very much for the review our paper.